# Teaming up main group metals with metallic iron to boost hydrogenation catalysis

Christian Färber[1], Philipp Stegner[1], Ulrich Zenneck[1], Christian Knüpfer[1], Georg Bendt[2], Stephan Schulz [2] & Sjoerd Harder [1✉]

Hydrogenation of unsaturated bonds is a key step in both the fine and petrochemical industries. Homogeneous and heterogeneous catalysts are historically based on noble group 9 and 10 metals. Increasing awareness of sustainability drives the replacement of costly, and often harmful, precious metals by abundant $3d$-metals or even main group metals. Although not as efficient as noble transition metals, metallic barium was recently found to be a versatile hydrogenation catalyst. Here we show that addition of finely divided $Fe^0$, which itself is a poor hydrogenation catalyst, boosts activities of $Ba^0$ by several orders of magnitude, enabling rapid hydrogenation of alkynes, imines, challenging multi-substituted alkenes and non-activated arenes. Metallic $Fe^0$ also boosts the activity of soluble early main group metal hydride catalysts, or precursors thereto. This synergy originates from cooperativity between a homogeneous, highly reactive, polar main group metal hydride complex and a heterogeneous $Fe^0$ surface that is responsible for substrate activation.

[1] Inorganic Chemistry, Friedrich-Alexander-Universität Erlangen-Nürnberg, Egerlandstrasse 1, 91058 Erlangen, Germany. [2] Institute of Inorganic Chemistry and Center for Nanointegration Duisburg-Essen (CENIDE), Universität Duisburg-Essen, Universitätsstrasse 5-7, 45141 Essen, Germany. ✉email: sjoerd.harder@fau.de

The hydrogenation of unsaturated molecules is a major achievement that pioneered the beginnings of heterogeneous[1] as well as homogeneous[2] catalysis. Despite its long history, hydrogenation catalysis is by no means old hat (Searching for "hydrogenation catalyst" in SciFinder gave a total of circa 33.000 hits of which nearly 11.000 date from the last decade (2010–2020) but today still a focal point of industrial as well as academic research[3]. Contemporary research is directed towards extending industrial applications, the conversion of biomass, and efforts to increase sustainability by replacing noble metals in classical hydrogenation catalysts with abundant metals[4].

Heterogeneous alkene hydrogenation catalysts are traditionally based on late transition metals (Pt/γ-Al$_2$O$_3$, Pd/C, or Raney-Ni)[5]. Being robust, these catalysts withstand the forcing conditions needed for the reduction of highly challenging substrates such as aromatic rings. Homogenous alkene hydrogenation catalysis initially focused on the metal Rh, with Wilkinson's catalyst RhCl(PPh$_3$)$_3$ as a well-known textbook example[6]. Further developments include Crabtree's cationic Ir complex which still today is one of the rare catalysts that is able to reduce most challenging tetrasubstituted alkenes like Me$_2$C$=$CMe$_2$ (Fig. 1)[7]. Although molecular catalysts are thermally sensitive and often need sophisticated ligands, the milder reaction conditions allow for higher selectivities.

Classical transition metal alkene hydrogenation follows two different routes: a dihydride pathway, which involves 2$e$ oxidation and reduction processes, or a monohydride cycle in which the metal oxidation state is unaffected (Fig. 2a). Current research on alkene hydrogenation includes early/late heterobimetallic cooperativity[8], ligand-metal cooperativity[9], cooperative hydrogen-atom-transfer ($c$HAT)[10] which may be combined with photoactivation[11], or Frustrated-Lewis-Pair (FLP) catalysis[12].

While the latter FLP protocol is a metal-free route for alkene hydrogenation, there is also a strong drive to develop hydrogenation catalysts based on broadly available, abundant metals[4,13]. This development is not limited to the transition metals but also sparked the rapidly growing area of early main group metal catalysis[14,15]. Access to the first soluble calcium hydride complexes[16] has led to a protocol for alkene hydrogenation under relatively mild conditions[17]. As calcium is not known for facile reversible redox reactions, this catalytic conversion follows the classical monohydride cycle (Fig. 2a). Since this initial report, there has been a considerable improvement in the performance of Ae metal catalysts[18–24]; Ae = alkaline earth metal. We demonstrated that common amide precursors like AeN″$_2$ (N″ = N(SiMe$_3$)$_2$) are highly effective precatalysts for hydrogenation of imines[21] and alkenes[22]. Catalyst initiation requires the unexpected deprotonation of H$_2$ (p$K_a$ ≈ 49) by the weak AeN″$_2$ base (p$K_a$ HN(SiMe$_3$)$_2$ = 25.8); Fig. 2b[25,26]. This apparent contrathermodynamic reaction is enabled by subsequent exothermic aggregation of Ae(H)N″ and AeH$_2$ species to give a variety of larger clusters for which examples have been isolated and structurally characterized, *cf.* [Ba(H)N″]$_7$ (Fig. 2b)[27,28]. Such mixed amide-hydride clusters of general formula Ae$_x$H$_y$N″$_z$ are thermally highly robust, enabling homogeneous catalysis in the 120–140 °C range with activities that increase going down group 2: Ca < Sr < Ba. Larger amide ligands led to smaller, more reactive clusters, considerably improving

catalyst activities. Using the precatalyst Ba[N(Si$i$Pr$_3$)$_2$]$_2$ extended the substrate scope to challenging tetrasubstituted alkenes and, although very slow, also benzene could be hydrogenated[23]. Considering that ligand bulk is favorable for high activities, the most recent observation that barium metal alone is an even better hydrogenation catalyst was unexpected[24]. The metal was activated by the evaporation/condensation method (metal vapor synthesis = MVS) producing a finely divided Ba$^0$ powder (MVS-Ba$^0$) which is highly reactive. This property is also utilized in radio tubes in which Ba$^0$ mirrors serve as a getter for various gasses[29].

The pathway for alkene hydrogenation with MVS-Ba$^0$ has been reported previously[24]. Although the reaction of the highly electropositive metal Ba$^0$ with H$_2$ normally needs a higher temperature (>80 °C)[30], it was shown that MVS-activated Ba$^0$ is already converted at room temperature[24]. However, full conversion to BaH$_2$ is never achieved and generally, substoichiometric compounds are formed[31]. The Ba$^0$/BaH$_2$ mixture is the starting point for two catalytic cycles (Fig. 2c). (1) The barium hydride cycle: alkene substrate can react with BaH$_2$, producing soluble Ba$_x$(H)$_y$R$_z$ clusters with highly reactive Ba-H bonds. Further alkene insertion is followed by hydrogenolysis, leading to alkanes and the reformation of barium hydrides. (2) The barium metal cycle: alkenes with conjugated (activated) C$=$C bonds can react with Ba$^0$ by oxidative addition, giving metallacycles. In addition to numerous examples[32,33], we demonstrated the oxidative addition of MVS-activated Ba$^0$ to Ph$_2$C$=$CPh2 or Ph$_2$C$=$NPh[24]. We also showed that the intermediates, [Ba$^{2+}$][Ph$_2$C-CPh$_2$$^{2-}$] or [Ba$^{2+}$][Ph$_2$C-NPh$^{2-}$], react with H$_2$ to the final hydrogenation products.

The concurrent existence of Ba$^0$ and Ba hydride species is essential for the unexpectedly high catalytic activity of MVS-Ba$^0$. This working hypothesis is based on Wright and Weller's early studies on ethylene hydrogenation with metallic Ca$^0$/CaH$_2$ or Ba$^0$/BaH$_2$ mixtures[34–36]. Corroborated by extensive experimental work, it was claimed that catalysis occurs at the interface between free metal and metal hydride: the Ba$^0$ surface activates alkenes for nucleophilic attack by polar Ba hydrides (Fig. 2d). This so-called dual-site mechanism is supported by experimental verification of ethylene adsorption on a Ba$^0$ surface[37]. Substrate activation by $d \rightarrow \pi^*$ backbonding from heavier Ae metals (Ca, Sr, Ba) is gradually gaining ground, especially for metals in low oxidation states[38–40]. This is exemplified by several recent observations like N$_2$ activation with Ca$^I$[41], considerable red-shifting of the CO stretching frequency in Ba(CO)$_8$[38], and benzene activation in Ba(benzene)$_3$[42]. π-Backbonding from $d$-orbitals on Ba$^0$ to benzene π*-orbitals results in pronounced electron transfer from metal to benzene and an overall elongation of the C-C bonds. Although the debate on the relevance of $d$-orbitals for the heavier Ae metals is currently highly controversial[39], it is notable that some overlap of $s$-, $p$- and $d$-bands in heavier Ae metals had already been discussed by Wright and Weller in the early 1950s[34].

Building upon this hypothesis, we now combine reactive main group metal hydride species with a transition metal surface. The rationale behind this idea is the fact that, in contrast to main group metals, transition metal surfaces are well known for their capability to activate unsaturated bonds. This concept of cooperative catalysis merges homogeneous catalysis, in the form of a

**Fig. 1 Catalytic alkene hydrogenation.** The Crabtree catalyst and the general rate order in alkene hydrogenation[52]. Activation of the C=C bond by conjugation with alkenes or arenes facilitates hydrogenation.

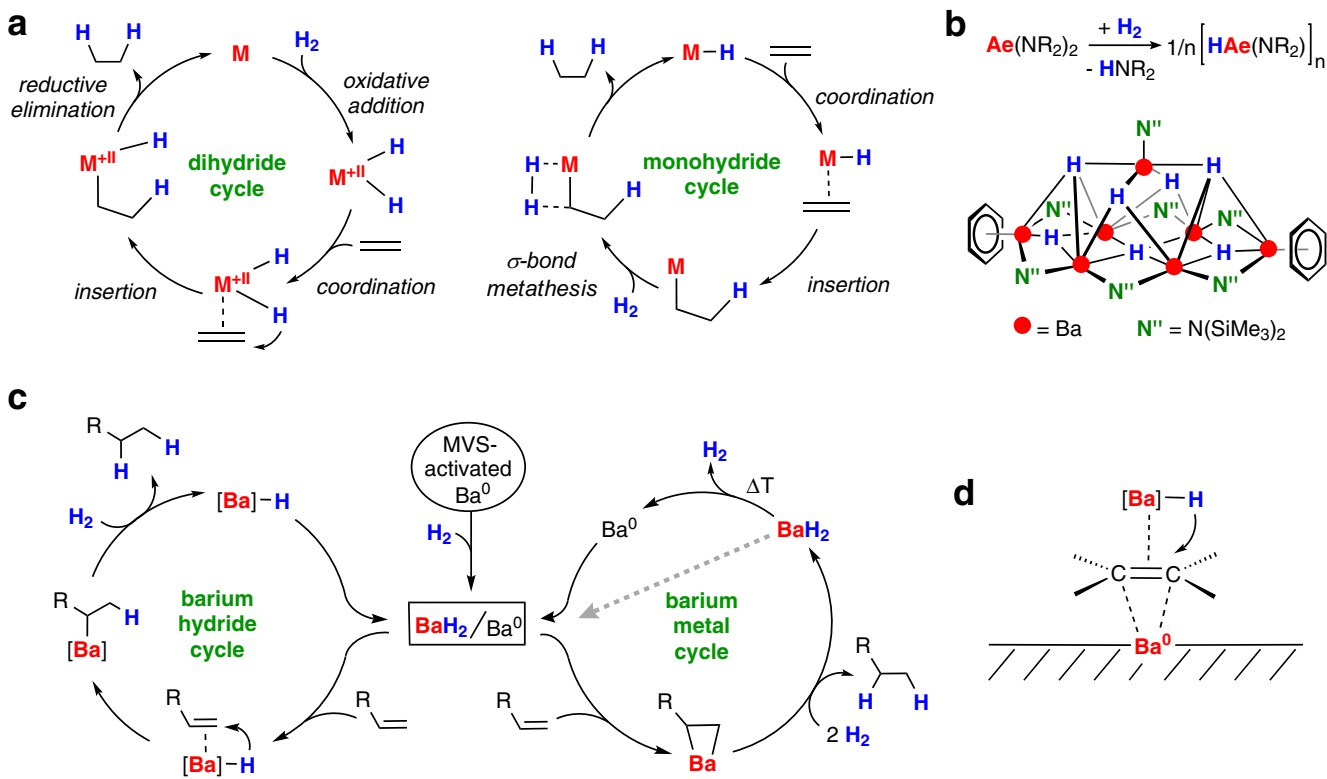

**Fig. 2 General mechanisms for alkene hydrogenation. a** The two fundamental cycles in transition metal-catalyzed alkene hydrogenation. **b** Formation of [Ae(H)NR$_2$]$_n$ aggregates (Ae = Ca, Sr, Ba) for catalytic alkene hydrogenation and an example of a structurally characterized [Ba(H)N″(C$_6$H$_6$)$_2$]$_7$ aggregate. **c** The two possible catalytic cycles for alkene hydrogenation with activated barium metal. **d** Proposed dual-site mechanism for Ba$^0$ promoted alkene hydrogenation.

soluble main group component, with heterogeneous catalysis, represented by the insoluble transition metal. For the latter, we chose iron which, as one of the most abundant metals with high biocompatibility, has been a focal point in contemporary homogeneous or heterogeneous hydrogenation catalysis[43–48] and plays a leading role in biochemical hydrogenation[49]. We demonstrate that a combination of early main group metal hydrides with a Fe$^0$ surface results in a hydrogenation catalyst that is considerably more active than any main group or iron catalyst alone.

### Results and discussion

**Catalyst preparation and hydrogenation catalysis**. MVS-activated Ba$^0$ was obtained in multi-gram quantities by cocondensation of barium metal and *n*-heptane[24]. Using a similar method, multi-gram quantities of a finely divided form of Fe$^0$ could be obtained. Cocondensation of Fe and toluene gave a red solution of the highly labile complex Fe(toluene)$_2$[50]. Controlled decomposition above −60 °C led to the formation of a black suspension from which a fine black Fe$^0$ powder was isolated.

Since no stabilizing organic capping agents were used, the MVS-activated Fe$^0$ and Ba$^0$ powders are extremely air-sensitive and highly pyrophoric. Elemental analysis and powder X-ray diffraction in sealed capillaries confirm their metallic state. The diffraction patterns show that both samples consist of microcrystalline nanoparticles of circa 5 nm (Fig. S1). While MVS-Fe$^0$ crystallized in a body-centered cubic (*bcc*) lattice typical for α-Fe, MVS-Ba$^0$ was obtained in a face-centered cubic (*fcc*) lattice. The latter β-Ba modification should be considered a metastable phase which was first observed by spraying barium vapor in a high vacuum, i.e., conditions similar to cocondensation by MVS[51].

Scanning-electron-microscope (SEM) and transmission-electron-microscope (TEM) studies show the formation of large agglomerates which are highly porous and consist of nanoparticles in the sub-10 nm regime (Figs. S2, 3). The metal particles are persistent to dispersion and extremely pyrophoric, characteristics that both prevent more accurate microscopy studies. X-ray photoelectron spectroscopy shows that the surface of these highly pyrophoric powders is partially oxidized due to sample preparation (Figs. S4–8).

Similar to previously reported Fe$^0$ nanoparticles[47,48], the MVS-activated Fe$^0$ powder catalyzed the hydrogenation of 1-hexene and cyclic di-substituted alkenes like cyclohexene (Fig. 3 and Table S3). It hardly reduced linear internal alkenes like 3-hexene and is inactive for the reduction of tri-substituted alkenes or arenes. In contrast, homogenous Fe complexes or clusters have been reported to reduce tetrasubstituted alkenes but are inactive in arene reduction[45,46]. MVS-activated Ba$^0$, which reduced benzene very slowly[24], is clearly superior to the herein obtained Fe$^0$ but not at par with noble *d*-block metal hydrogenation catalysts. We now found that an equimolar Ba$^0$/Fe$^0$ mixture is up to three orders of magnitude more active than the most active Ba$^0$ component alone (Fig. 3). This allows facile hydrogenation with very low catalyst loadings and further extension of the substrate scope to the most challenging arene substrates.

Using a very low catalyst loading of only 0.05 mol% BaFe, 1-hexene was fully reduced within 15 min. The addition of Fe$^0$ to Ba$^0$ led to an increase in the turn-over-frequency (TOF) from 7 h$^{-1}$ (Ba) to 8000 h$^{-1}$ (BaFe), measured at full conversion. Also, internal alkenes were reduced efficiently with the expected reactivity order: cyclic alkenes > *cis*-alkenes > *trans*-alkenes > tri-substituted alkenes[52]. Hydrogenation of the unactivated C=C bond in 1-Me-cyclohexene is surprisingly fast (TOF =

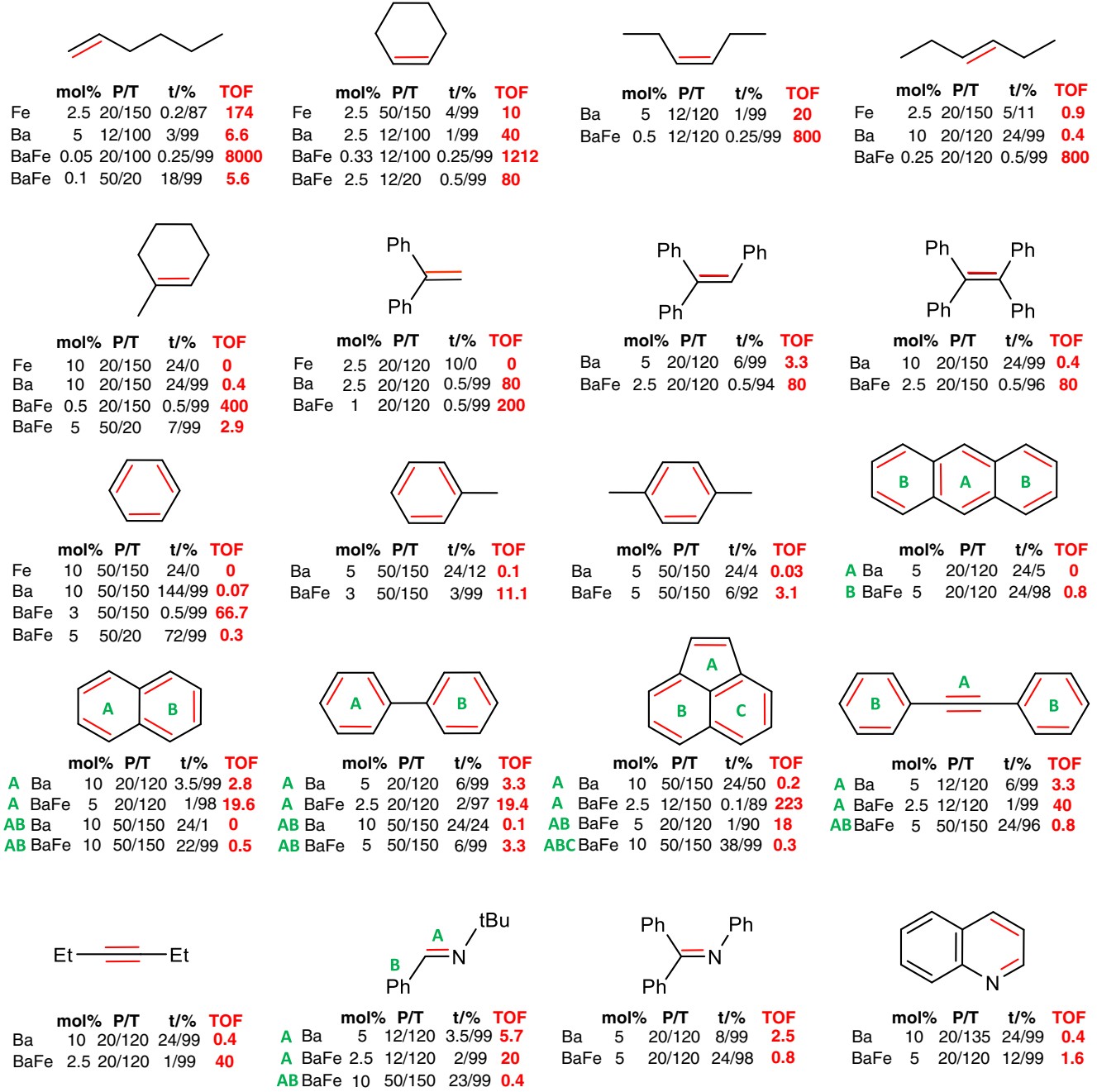

**Fig. 3 Catalytic hydrogenation.** Comparison of the activities for $Ba^0$ and the mixed $Ba^0/Fe^0$ catalysts (the mixture was ground before use). P = $H_2$ starting pressure in bar (reactors were pressurized with $H_2$ and shut off from the $H_2$ source), T = temperature in °C, and t = time in hours which in most cases has been optimized for full conversion (99%). For comparison, TOF values (turn-over-frequency in $h^{-1}$) are given. The controversial use of TOF values is discussed in the Supplementary Information.

400 $h^{-1}$). The BaFe mixture achieved even for tetrasubstituted $Ph_2C=CPh_2$ quantitative reduction within the hour. However, the slightly higher temperature also led to traces of Ph ring reduction, indicating that the BaFe catalyst is highly efficient for benzene-to-cyclohexane conversion. Homo- or heterogeneous Fe catalysts alone are fully inactive in arene hydrogenation[45–48], while for $Ba^0$ a high catalyst loading of 10 mol% and at least 6 days are needed for full benzene-to-cyclohexane conversion[24]. Under the same conditions, but with only 3 mol% catalyst loading, BaFe reduced benzene quantitatively within 0.5 h. Even electron-rich, alkylated arenes like toluene and p-xylene could be fully hydrogenated, with rates decreasing with the degree of alkylation.

The BaFe catalyst also hydrogenated polycyclic aromatic hydrocarbons and, depending on the conditions, selective reduction was obtained. Interestingly, whereas $Ba^0$ reduces the central ring in anthracene only stoichiometrically, the BaFe combination hydrogenates the terminal rings catalytically. This demonstrates that $Fe^0$ addition influences both, activity and selectivity. Naphthalene and biphenyl, both substrates in which reduction of one of the rings is facile, could be fully hydrogenated using BaFe. Whereas $Ba^0$ alone barely reduced the exposed double bond in acenaphthylene, the BaFe catalyst managed full conversion within minutes and, controlling the reaction conditions, all three rings could be reduced stepwise. Although ketones could not be hydrogenated, the scope of the BaFe catalyst was

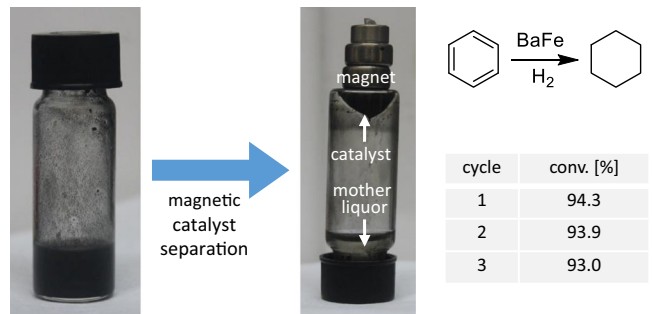

**Fig. 4 Catalyst recycling.** The BaFe catalyst after benzene-to-cyclohexane hydrogenation (2.5 mol%, 50 bar, 150 °C, 0.5 h) can be separated conveniently with a magnet and was recycled.

| cycle | conv. [%] |
|---|---|
| 1 | 94.3 |
| 2 | 93.9 |
| 3 | 93.0 |

**Table 1 Catalytic benzene hydrogenation with combined X/Fe catalysts (1.5 mol%, 150 °C, 50 bar, 2 h); X = main group metal species; N″ = N(SiMe₃)₂; BDI = HC[C(Me)N-DIPP]₂ (DIPP = 2,6-diisopropylphenyl).**

| X | conv. [%] | TOF [h$^{-1}$] |
|---|---|---|
| Ba$^0$ | >99 | >33 |
| CaH$_2$ | 0 | 0 |
| SrH$_2$ | 3 | 1 |
| BaH$_2$ | 82 | 21 |
| MgN″$_2$ | 32 | 11 |
| CaN″$_2$ | 29 | 10 |
| SrN″$_2$ | 40 | 13 |
| BaN″$_2$ | 86 | 29 |
| LiN″ | 2.5 | 1 |
| NaN″ | 10 | 3 |
| KN″ | 40 | 13 |
| [Ba(H)N″]$_7$ | 21 | 7 |
| [(BDI)MgH]$_2$ | >99 | >33 |

further extended to alkynes and imines in which under harsher conditions also the Ph substituents could be reduced. The heteroatom-containing ring in the Lewis base quinoline was fully hydrogenated within 12 h. The cooperative effect between Ba$^0$ and Fe$^0$ is for substrates with heteroatoms in general much smaller than for alkenes and arenes. This may originate from differences in substrate-surface bonding.

Like the precatalysts AeN″$_2$ and Ba$^0$ [22,24], the BaFe catalyst is extremely robust and easily tolerates temperatures up to 150 °C. However, with higher catalyst loading and longer reaction times full conversions can also be reached at room temperature (Fig. 3). The lower operation temperature allows catalytic runs in reactors that are open to the H$_2$ source. This shows that hydrogenation at a low constant pressure of 6 bar is equally effective (Table S4) and that high initial pressures are not needed. Not only the H$_2$ pressure but also the substrate concentration has no effect on the conversion rate (Table S5).

Investigations concerning the Ba$^0$/Fe$^0$ ratio revealed that the activity in benzene reduction increases linearly with Fe content, reaching an optimum at a ratio of circa 1/1 (Fig. S15). Mechanically grinding a mixture of both metal powders with mortar and pestle prior to catalysis, increased TOF's consistently by a factor of 10 (Table S11). This simple method of mixing results in a rather heterogeneous Ba$^0$/Fe$^0$ distribution (energy dispersive X-ray (EDX) mapping: Figs. S9–11). The powder X-ray diffraction (p-XRD) and X-ray photoelectron spectra (XPS) for the ground BaFe mixture is a superposition of Ba$^0$ and Fe$^0$ spectra (Figs. S1, S4). This is in agreement with the observation that Ba$^0$ and Fe$^0$ do not form alloys[53]. Considering that Ba$^0$ is soft (1.25 Mohs) and Fe$^0$ is hard (4.0 Mohs)[54], it is plausible that an intimate surface contact of both metals is beneficial for catalyst activity. Indeed, whereas a Ba$^0$/Fe$^0$ mixture can still be magnetically separated in its elements, the ground powder is fully magnetic and cannot be separated.

Not surprisingly, the activation of Fe$^0$ is an essential requirement for catalytic activity. Commercially available iron powder is not active but pyrophoric Fe$^0$, obtained by thermal decomposition of Fe-oxalate, also boosts the activity of Ba$^0$. It is, however, clearly inferior to MVS-activated Fe$^0$ (Table S7).

The heterogeneous nature of the initial Ba$^0$/Fe$^0$ mixture suggests that the catalyst may be recycled. Hydrogenation of benzene to cyclohexane with BaFe gave a black suspension from which the catalyst can be removed from the mother liquor with a magnet and reused without significant loss of activity (Fig. 4). The mother liquor does not contain dissolved salts or complexes (after evaporation of all volatiles no visible products remained, see Fig. S16). As $^1$H Nuclear-Magnetic-Resonance (NMR) spectra of the mother liquor do not show line-broadening, the presence of trace quantities of paramagnetic iron species is excluded.

**Catalyst variations**. The herein presented concept of cooperative main group/transition metal catalysis with a Ba$^0$/Fe$^0$ mixture can be extended to other Ae$^0$ metals. Although BaFe is one of the most active catalysts, activities for MgFe, CaFe, and SrFe are all within a similar order of magnitude (Tables S9, 10), indicating that also abundant, biocompatible, metals like Mg$^0$ and Ca$^0$ can be used successfully.

Assuming that the Ae$^0$ metal reacts with H$_2$ to form AeH$_2$, we investigated commercially available AeH$_2$ salts as a catalyst component. Whereas BaH$_2$ and Fe$^0$ are individually fully inactive for benzene reduction, its combination is a potent catalyst (Table 1). However, the mixtures CaH$_2$/Fe$^0$ and SrH$_2$/Fe$^0$ were essentially inactive. This may be related to the high lattice energies and poor solubilities of these commercially available metal hydrides. Metal amides, AeN″$_2$ (Ae = Mg, Ca, Sr, Ba), which under H$_2$ atmosphere convert to soluble hydride clusters[27,28], are also effective cocatalysts for benzene hydrogenation. On their own, these amides cannot reduce benzene but the AeN″$_2$/Fe$^0$ combination is especially for Ba quite active (Table 1). The corresponding alkali metal amides gain activity with increasing metal size (LiN″ < NaN″ < KN″) but are in general less active than AeN″$_2$. Interestingly, while the hydrogenation of unactivated Me2C=CMe2 could not be achieved with BaFe, this most challenging substrate was fully reduced with the MgN″$_2$/Fe$^0$ combination (3 mol%, 150 °C, 50 bar, 6 h, TOF = 6 h$^{-1}$); Table S6. The Fe$^0$ additive turns MgN″$_2$, which thus far has not shown any activity for hydrogenation of even the simplest alkenes[14,15,22], into a very potent hydrogenation catalyst.

The assumption that Ae$^0$ or AeH$_2$ form under catalytic conditions a soluble metal hydride species is supported by the fact that Ba$^0$ in the BaFe catalyst can be replaced by the soluble hydride cluster [Ba(H)N″]7 (Fig. 2b). As the fate of the [Ba(H)N″]7 cluster under catalytic conditions is unclear, this is not ultimate proof for a soluble main group metal component. However, we also found that the combination of [(BDI)MgH]$_2$ and Fe$^0$ is a highly active catalyst for benzene hydrogenation (Table 1 and Fig. 5). The robust complex [(BDI)MgH]$_2$ could be seen as magnesium hydride which has been solubilized by a bulky β-diketiminate (BDI) ligand[55]. As a stand-alone catalyst, [(BDI)MgH]$_2$ is fully inactive in benzene hydrogenation (10 mol%, 50 bar H$_2$, 150 °C) but in combination with MVS-Fe$^0$ it is competitive with Ba$^0$/Fe$^0$. After magnetic separation of the Fe$^0$ cocatalyst, [(BDI)MgH]$_2$ was the only complex that could be detected by $^1$H NMR (Figs. S12–14). Based on these observations,

**Fig. 5 Proposed mechanism for cooperative benzene hydrogenation.** Benzene, activated at a $Fe^0$ surface, is hydrogenated by a soluble Mg hydride complex.

it is likely that catalysis proceeds at the solid-solution interface. We propose a mechanism in which the soluble, homogenous Mg hydride catalyst works in concert with an insoluble, heterogenous $Fe^0$ catalyst (Fig. 5). The recognition that main group metal catalysts are activated by $Fe^0$ seems therefore a general principle that opens up numerous possibilities for future research.

**Mechanistic considerations.** Although there is compelling evidence that $Fe^0$ and $Ba^0$ act in synergy in catalytic alkene hydrogenation, it is difficult to fully reveal the intricate details of the mechanism. Being insoluble in organics, the BaFe precatalyst is clearly a heterogeneous system. As the catalyst can be fully recycled without loss of activity (Fig. 4), and assuming that catalytic activity is not due to undetected trace metal quantities in solution, it is also after catalysis of a heterogeneous system. A tentative mechanism for what happens during catalysis can only be postulated based on observations, reported literature and common knowledge.

p-XRD and XPS investigations on the BaFe catalyst, obtained by grinding MVS-activated $Ba^0$ and $Fe^0$ powders, show a rather inhomogeneous element distribution. For this reason, the mechanism previously reported for $Ba^0$/$BaH_2$ catalyzed alkene hydrogenation (Fig. 2c) would be a logical starting point. Indeed, as previously shown for $Ba^0$ catalysis[24], we here show that a spent BaFe catalyst contains metallic $Ba^0$ and hydride functions (Figs. S17–S20). Since [1]H NMR spectra of the mother liquor after catalyst separation show no sign of line broadening due to contamination with paramagnetic Fe species, it is likely that $Fe^0$ remains heterogeneous during catalysis. Addition of liquid mercury has a poisoning effect on catalyst activity (Table S12), hinting that under catalytic conditions there is indeed a heterogeneous catalyst component[56]. Another indication of a catalytic reaction in which the key steps are heterogeneous comes from the observation that conversion rates are independent of $H_2$ pressure and substrate concentration. This stands in strong contrast with homogeneous Ae metal-catalyzed alkene hydrogenation in which higher $H_2$ pressure is beneficial[14,17] and, in agreement with calculations[57], hydrogenolysis by σ-bond metathesis is rate-determining (Fig. 2a). The pseudo-zero order behavior in BaFe catalyzed alkene hydrogenation is typical for a heterogeneous reaction in which substrate-active site interaction is the rate-limiting factor.

Although the catalytic reaction has a heterogeneous character, there is evidence that the main group component is during catalysis at least partially in solution. This is demonstrated by the fact that $Fe^0$ also boosts the catalytic activity of soluble Ae metal hydrides or precursors thereto (Table 1). Although these examples may not be pertinent to the BaFe system, EDX-mapping of a fresh BaFe and a spent BaFe catalyst provide strong evidence for a soluble Ba component. While the surface of a fresh catalyst mixture shows an expected Ba/Fe ratio of circa 1/1 (Fig. S9 and Table S1), the surface of a spent BaFe catalyst is considerably enriched in Ba with a Ba/Fe ratio of circa 2/1 (Fig. S10 and Table S2). Since the catalyst is after catalysis fully recovered, this observation suggests that Ba is solubilized during catalysis and precipitates after full substrate conversion, either as $Ba^0$ or $BaH_2$.

Building upon the mechanism previously discussed for hydrogenation with $Ba^0$ (Fig. 2c)[24] and comprehensive reports by Wright and Weller on $Ae^0$/$AeH_2$ catalysts[34–36], we postulate for the $Ba^0$/$Fe^0$ mixture a mechanism in which homogeneous Ba and heterogeneous $Fe^0$ catalysts work in synergy (Fig. 6). MVS-Activated $Ba^0$ reacts, dependent on the nature of the substrate, either first with $H_2$ or with an alkene to give a highly reactive, solubilized Ba hydride species. The cycle is closed by alkene insertion followed by hydrogenolysis with $H_2$. There are several steps (marked A–E in Fig. 6) in which the heterogeneous $Fe^0$ catalyst could play an activating role:

(A) Partially filled *d*-orbitals on $Fe^0$ facilitate H-H bond cleavage by backbonding in the σ* orbital (LUMO). Dissociative adsorption of $H_2$ on an Fe(110) surface has been calculated to be barrier-free[58]. This results in atomistic hydrogen which can move freely in the iron crystal lattice and is the origin of hydrogen embrittlement of steel[59]. The presence of highly reactive H radicals could facilitate the $Ba^0 + 2 H \rightarrow BaH_2$ conversion, resulting in a salt composed of $Ba^{2+}$ and $H^-$ ions. In situ generated, unaggregated $BaH_2$ will be a very potent reducing agent, reacting smoothly with alkenes to give an R-Ba-H species which, similar to $[Ba(H)N'']_7$[28], can go into solution in the form of a $[Ba(H)R]_n$ aggregate.

(B) The $Fe^0$ surface can activate unsaturated substrates for hydride attack, an ability that is strongly supported by theory. It is generally agreed that in hydrogenation over a group 8 metal catalyst, benzene is associatively adsorbed as a π-complex[60]. Calculated benzene adsorption energies on a Fe(110) surface are comparable to those for adsorption at (111)-surfaces of Ni, Pd or Pt[61]. Similar as in molecular metal-benzene complexes, the $Fe^0$ surface accepts π-electrons from benzene's π-HOMO and donates electrons back in the π*-LUMO[62]. Concomitant C-C bond lengthening and decrease of the HOMO-LUMO gap facilitate hydride attack.

(C) Highly reactive H radicals that are formed at the $Fe^0$ surface could directly react with alkylbarium intermediates to form the final product. This would facilitate the hydrogenolysis step, *i.e.* the σ-bond metathesis step which under homogeneous conditions is rate-determining[14,17,57].

(D) As previously described for $Ae^0$/$AeH_2$ systems[24,34–36], desorption of $H_2$ from $BaH_2$ starts at temperatures above 100 °C. Since it is well-known that $Fe^0$ nanoparticles catalyze the hydrogen desorption kinetics for the $Mg^0$/$MgH_2$ system[63], the elimination of $H_2$ from $BaH_2$ could be facilitated by MVS-activated $Fe^0$. Note that this step is only required for a full $Ba^0$ cycle.

(E) Metal mixtures of $Fe^0$ and $Ba^0$ could profit from an additional activating effect. Comparable with K-promotion of the

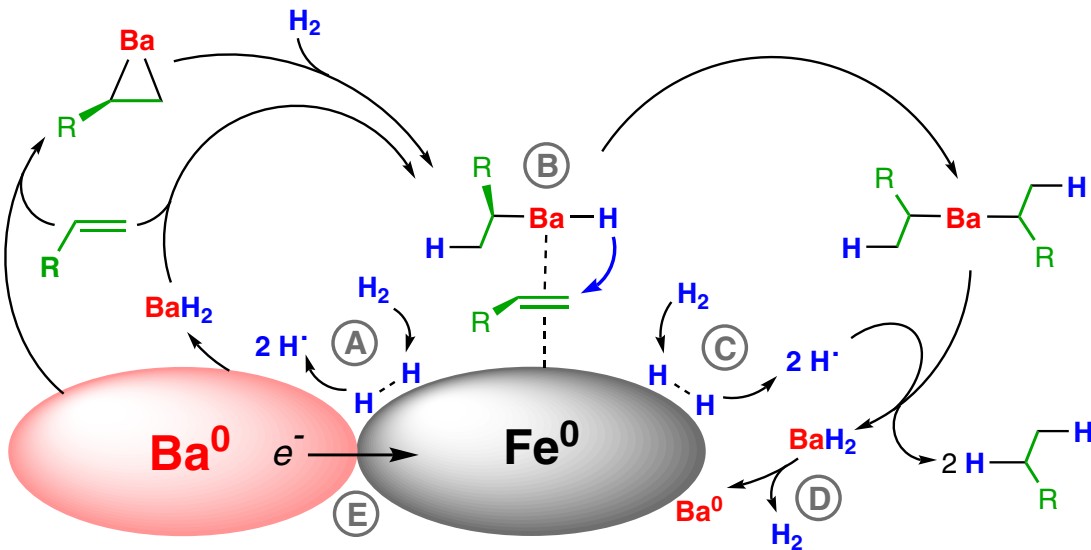

**Fig. 6 Synergy between $Ba^0$ and $Fe^0$ in catalytic alkene hydrogenation.** The Ba component enters the solution phase by reaction of electropositive $Ba^0$ with $H_2$ or alkene. The following steps are potentially supported by the heterogeneous $Fe^0$ component: (A) $Fe^0$ facilitates H-H bond cleavage and $BaH_2$ formation. (B) Alkene activation by $Fe(d) \rightarrow \pi^*$(alkene) backdonation. (C) Hydrogenolysis of alkylbarium intermediates is facilitated by the presence of H radicals. (D) $Fe^0$ catalyzes the desorption of $H_2$ from $BaH_2$. (E) Ba-Promotion of $Fe^0$ by electron injection.

Haber-Bosch Fe-catalyst for $N_2$ activation[64], electropositive $Ba^0$ may enforce the *e*-transfer chain $Ba^0 \rightarrow Fe^0(d) \rightarrow \pi^*$(alkene), giving substrate activation a supplementary boost. Most recently, a Cs-promoted Fe catalyst enabled $N_2$-to-$NH_3$ conversion at room temperature[65]. It has been reported that Ba-promotion can be even more effective than Cs-promotion[66]. Such promotion requires an intimate BaFe contact and is in agreement with the herein described increase of catalytic activity by mechanical mixing of $Ba^0$ and $Fe^0$.

While we are further scrutinizing the intricate details of this tentative mechanism, the hitherto made observations have important ramifications for future research. The proposed mechanism suggests that the main group metal component can switch between solution and solid states, while the $Fe^0$ catalyst remains heterogeneous. There is currently no indication that trace quantities of solubilized Fe are responsible for catalytic activity, however, this cannot be fully excluded. Even in explicit cases of homo- or heterogeneous catalysis, the "true" catalytic species can be a cocktail of molecular catalysts and nanoparticles[67]. We are currently exploring possibilities to convert the concept to a fully homogenous system which will facilitate further mechanistic studies.

### Concluding remarks

We demonstrated that a mixture of $Ba^0$ and $Fe^0$ is a highly potent hydrogenation catalyst featuring activities that considerably surpass those of their individual components. This synergy expands their substrate scope to challenging arenes, raising the level to that known for the more noble group 9 or 10 metal catalysts. This heterobimetallic strategy could be extended to other Ae metals, including abundant, biocompatible metals like $Mg^0$ and $Ca^0$. A further demonstration of the sustainability of this concept is demonstrated by facile magnetic separation and recycling of the $Ba^0/Fe^0$ catalyst without significant activity loss.

The assumption that the main group component in $Ae^0/Fe^0$ forms a soluble Ae metal hydride species, is supported by the observation that $Fe^0$ also boosts the catalytic activity of early main group metal hydride complexes (or precursors thereto). A striking example of the effectiveness of this cooperative catalytic system is

the facile benzene-to-cyclohexane conversion by a mixture of $[(BDI)MgH]_2$, a soluble well-defined Mg hydride complex, and $Fe^0$.

Although it is difficult to gain insight in processes at the interface of homogeneous and heterogeneous catalysis, experimental observations suggest that the spent BaFe catalyst is a mixture of $Fe^0$, $Ba^0$, and hydride species. Based upon previous work and current data, we propose a mechanism in which $Fe^0$ plays an activating role in H-H bond cleavage and $BaH_2$ formation but also activates the unsaturated substrate at its surface by $\pi$-backbonding. The $Ba^0$ component is the source for soluble Ba hydride reagents and also may play a role as an electronic promoter for $Fe^0$.

Boosting the activities of homogeneous main group metal catalysts by interplay with a $Fe^0$ surface merges homogeneous with heterogeneous catalysis. The simplicity of this concept and the numerous possibilities for metal combinations, hold considerable potential in the future search for sustainable hydrogenation catalysts based on abundant metals.

### Data availability

Supplementary information (general experimental procedures, experimental details for catalyst preparation and catalysis, NMR spectra, investigations towards the nature of the catalyst) is available within the Supplementary Material files, or from the corresponding author on request.

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

## Acknowledgements

We thank Samuel Grams for GC-MS measurement and Antigone Roth for elemental analysis. The Friedrich-Alexander-University Erlangen-Nürnberg is acknowledged for generous support.

## Author contributions

C.F.: Conceptualization, investigation, validation, formal analysis, writing—original draft, and visualization. P.S., C.K., and G.B.: Investigation, validation, and formal analysis. U.Z.: Conceptualization, investigation, validation, and formal analysis. S.S.: Writing—original draft—review and editing, visualization, and supervision. S.H.: Conceptualization, writing—original draft —review and editing, visualization, supervision, and project administration.

## Funding

## Competing interests

The authors declare no competing interests.
