## [Peer Review File · Nature Communications]

REVIEWER COMMENTS

Reviewer #1 (Remarks to the Author):

Herein, the authors expand upon their previous results concerning a Ba-based hydrogenation protocol and describe the preparation, characterization, and catalytic application of an augmented BaFe composite material that effects the hydrogenation of (bulky) alkenes, (annulated) arenes, alkynes, imines, and parent quinoline. A nice synthetic feature of the given catalyst is the control of regio- and/or chemoselectivity by variation of the catalyst loading and the physical reaction parameters. Furthermore, the spent catalyst is magnetically separated from the reaction mixture which makes time-consuming centrifugation steps superfluous. The manuscript is well-written and presentation of the scientific data is comprehensive. Given the high activity and robustness of this alkaline-earth-based hydrogenation method I am convinced that this paper will attract the interest of the readers of Nature Communications and thus this reviewer recommends the publication of the submission in the given journal.

Minor comments:

(1) Manuscript title: I guess the authors mean 'Teaming up ...' rather than 'Teaming ...'.

(2) Abstract and later in the text (page 2, page 7): 'non-activated' arenes instead of 'unactivated'.

(3) page 7: 'bulk-producing hydrides'

(4) page 8, Figure 5 (caption): (E) 'Hydrogenolysis' and not 'Hydrogenolysis'

(5) page 9: 'Concomitant' instead of 'Concomittant'

Reviewer #2 (Remarks to the Author):

The manuscript by Faerber et al presents a very interesting investigation into the synergistic effect of Fe combined with Ba for double bond hydrogenation. The results are sound and I enjoyed and appreciated the manuscript.

While performance part of the work appears solid to me, i have several concerns that need resolution before the publication relating specifically to the analysis of the catalytic action.

A) Mechanistic analysis:

This is my major concern. Authors build their mechanistic proposal largely on existing precedents (hence lacking novelty to some extent) without performing any experimentation targeted at understanding their specific catalyst system.

While the literature can back many claims and make them appear sensible -sensitivity does not imply relevance. In current form, the majority of arguments lack relevance to catalyst in question as they examine some other catalytic systems.

As a result, the proposal in its current form is a collection of options with no way of limiting their number. This somewhat limits the usefulness of analysis and justification for its inclusion.

I invite the authors to perform additional experiments to clarify what makes this excellent catalytic system so special. Specifically, authors might opt at performing reaction kinetics measurements and identify the RDS in Ba vs BaFe catalyzed reduction. Similarly, comparative KIE analysis might explain why combining Ba and Fe gives the performance such a boost.

Below are some follow up points on the mechanistic part:

1) Line 194: The claim is not correct or valid in isolation. The fact that addition of soluble barium instead of metallic one promotes catalysis is not relevant to the catalytic system in question. It doesnt prove that homogeneous species are in fact produced when using Ba/Fe, at least not in current wording.

2) Lines 207-208: This is very impressive.

3) Point A: Could authors clarify the state of hydrogen in each case - atomic vs hydride?

3) Point C is not demonstrated experimentally in the context of this work

4) Point D demonstrates that Fe can promote LAH reduction (this was noted by other referees) but is does not have direct relevance to the Fe-Ba catalyst in question

5) Point F is much appreciated. This is an exemplary, thoroughly analyzed piece of data that is directly relevant to catalysis in question

B) State and composition of the catalyst:

1) Elemental composition: can authors truly call their samples bare? EA evidences significant amount of carbon and nitrogen in extent of few percent and no mention of that is given in XPS data analysis. What is the role of these components? Do they affect catalysis?

2) Mechanical agitation: This is a very interesting observation and authors might follow up on why mechanical treatment impacts the catalytic activity. It is very unexpected and I urge you to address this. Firstly, this is critical for reproducibility. Secondly, does the electronic structure of the mixed catalyst change upon mechanical treatment?

3) Authors are encouraged to be very specific with respect to their numbers. In the current SI it is very unclear which amounts of catalysts were used especially in mixed catalyst systems and ones with alkali metal hydrides. In some instances the numbers are simply not there.

C) Further remarks in order of occurrence:

1) Can the authors clarify the LAH and DIBAL data? These are significantly different from Ae co-catalysts and might operate via a different mechanism, e.g. they show activity in the absence of H₂. In current state it appears you cannot reliably group these with Ae species. A control experiment would help clarifying this - e.g. one in the absence of H₂ and one without aqueous workup.

2) lines 35-40 - a more exhaustive description of homogeneous catalysis for alkene hydrogenation is needed for a broad readership journal

3) Figure 1 - a reference to the reactivity order is missing

4) Lines 47-50 - while i agree with the introduction of two mechanisms, the examples of redox non-active processes are very limited and somewhat insular. There are a lot of homogeneous (cooperative/bifunctional) catalyst that operate without redox and are not mentioned

5) Authors might reconsider their amide notation. The N" is actually HMDS - an exhaustive 4-letter notation that is clear to the majority of organic chemists

6) Lines 67-68: Could you elaborate on the nature of BaH₂ and give the reference to the mentioned reactivity?

7) Line 119: Could you elaborate on your use of the term "activated"?

8) Figure 3 and the manuscript: As long as there is no kinetic data, authors cannot use the term TOF, which is a kinetic performance parameter and cannot be estimated from single point. This would be strongly misleading and unjustified.

9) Figure 3 caption: typo in the word "shut"

10) Line 167 and throughout the manuscript: Authors need to clarify the weight vs surface aspect of their catalytic system. Since the catalyst is heterogeneous, how do weight ratios translate to surface ones? This point appears to be important since even mechanical treatment of pristine powder mixtures does strongly affect the performance (Line 170)

11) Line 179:

SI:

1) The accuracy of MVS synthesis description is critical for the reproducibility of results. Currently this description contains, i quote: "rule of thumb". How are other researchers supposed to reproduce this?

2) The protocol for Fe oxalate decomposition is missing and should be described

3) SI and line 179 - the data on elemental analysis of spent catalytic medium is not included. Without it and adequate detection limit description the claim on "quantitative" removal is unsupported

Summary: I find the results very appealing and invite the authors to act on the comments above. Once these issues are resolved, i'd be happy to recommend the publication.

Point-by-point response to the reviewers' comments

Reviewer #1

Herein, the authors expand upon their previous results concerning a Ba-based hydrogenation protocol and describe the preparation, characterization, and catalytic application of an augmented BaFe composite material that effects the hydrogenation of (bulky) alkenes, (annulated) arenes, alkynes, imines, and parent quinoline. A nice synthetic feature of the given catalyst is the control of regio- and/or chemoselectivity by variation of the catalyst loading and the physical reaction parameters. Furthermore, the spent catalyst is magnetically separated from the reaction mixture which makes time-consuming centrifugation steps superfluous. The manuscript is well-written and presentation of the scientific data is comprehensive. Given the high activity and robustness of this alkaline-earth-based hydrogenation method I am convinced that this paper will attract the interest of the readers of Nature Communications and thus this reviewer recommends the publication of the submission in the given journal.

Minor comments:

- (1) Manuscript title: I guess the authors mean 'Teaming up ...' rather than 'Teaming ...'.
- (2) Abstract and later in the text (page 2, page 7): 'non-activated' arenes instead of 'unactivated'.
- (3) page 7: 'bulk-producing hydrides'
- (4) page 8, Figure 5 (caption): (E) 'Hydrogenolysis' and not 'Hydrogenolysis'
- (5) page 9: 'Concomitant' instead of 'Concomittant'

All typos/grammar errors have been corrected, except for "bulk-producing hydrides". This part relates to LiAlH_4 which has been taking out of the paper (see below).

Reviewer #2

The manuscript by Faerber et al presents a very interesting investigation into the synergistic effect of Fe combined with Ba for double bond hydrogenation. The results are sound and I enjoyed and appreciated the manuscript.

We thank the reviewer for his appreciation of our sound manuscript.

While performance part of the work appears solid to me, i have several concerns that need resolution before the publication relating specifically to the analysis of the catalytic action.

A) Mechanistic analysis:

This is my major concern. Authors build their mechanistic proposal largely on existing precedents (hence lacking novelty to some extent) without performing any experimentation targeted at understanding

their specific catalyst system.

There is indeed existing precedence for only-Fe⁰ or only-Ba⁰ catalysts. We do not agree with “*lacking novelty to some extent*”. The novelty of the work lies in the demonstration that the combination of Ba⁰ and Fe⁰ gives for a unique catalytic system that is several orders of magnitude (> 1000 times) better than the separate metals and in certain cases even better! There is no definitive proof but there are many observations supporting the intricate details of the proposed mechanism. Apart from that, the ramifications of such bimetallic catalysis are enormous. The possible combinations of main group and transition metals are numerous and we feel that we are only at the start of the beginning.

The proposed mechanism for our BaFe catalyst is a weak point of the paper but it is not completely fair to state that “*we didn't perform any experimentation targeted at understanding our specific catalyst system.*” After all, we did show the following:

- optimization of the Ba⁰/Fe⁰ ratio
- acceleration of catalysis by grinding of both metals
- H₂ pressure does not influence conversions, hence is not rate-determining
- other alkaline-earth metals (Mg, Ca, Sr) are also active in combination with Fe⁰
- soluble main group metal hydride complexes (or precursors thereto) are active in combination with Fe⁰
- powder X-ray diffraction data of Ba⁰, Fe⁰ and Ba⁰/Fe⁰ show microcrystalline particles of circa 5 nm
- TEM and SEM studies on Ba⁰, Fe⁰ and the Ba⁰/Fe⁰ mixture show larger aggregates composed of smaller (circa 5 nm) particles.
- XPS surface studies show a largely heterogeneous distribution of Ba⁰ and Fe⁰ (which after simple grinding with mortar and pestle is to be expected).
- since the catalyst can be fully recycled with a magnet (but is not magnetic itself) it consists of Fe⁰
- the Ba/Fe surface ratio of an unused Ba⁰/Fe⁰ catalyst is circa 1/1, whereas that in a spent catalyst is circa 2/1. This means that Ba⁰ goes into solution during catalysis and precipitates afterwards.
- the catalyst is poisoned by mercury and therefore at least partially heterogeneous.
- the Ba component in a spent Ba⁰/Fe⁰ catalyst consists of Ba⁰ and reducing Ba hydride species like BaH₂

Additional observations for the Ba⁰ catalyst (without Fe⁰) were reported previously (*Angew. Chem. Int. Ed.* **2021**, *60*, 4252):

- activated Ba⁰ reacts with H₂ to BaH₂ already at 20 °C.
- activated Ba⁰ can also reduce conjugated alkenes (*cf.* Ph₂C=CPh₂) or Ph₂C=NPh resulting in [Ba²⁺][Ph₂C-CPh₂²⁻] or [Ba²⁺][Ph₂C-NPh²⁻].

These combined observations led to the proposal of the mechanism. I like to underscore the word “proposal”. It is not written in stone and, like any mechanism, subject to discussion. We wrote: “Although subject to discussion, this working hypothesis

There are several reports in very high impact journals, also in the *Nature series*, in which new catalyst systems are far less understood and mechanisms were either not or very poorly discussed. Given the fact that mechanisms for heterogeneous catalysts are extremely hard to study, we came up with a fair proposal of how things could work. In our revised manuscript we now present even more convincing evidence (see below).

While the literature can back many claims and make them appear sensible -sensitivity does not imply relevance. In current form, the majority of arguments lack relevance to catalyst in question as they examine some other catalytic systems.

If I interpret this correctly, the reviewer says that the observations/explanations described in the quoted literature are not relevant for the Ba^0/Fe^0 and main-group metal hydride/ Fe^0 catalysts described in our paper (with the argument that they deal with other catalytic systems).

All analyses support a catalyst which is a mixture of nanocrystalline Ba^0 and Fe^0 (with a rather heterogeneous distribution). The two metals form no legation. The papers we quote discuss experiments or calculations directly related to interaction of Fe^0 or Ba^0 (our catalyst) with alkene/arene or H_2 (our educts).

In detail:

Point (A) in our (unrevised) manuscript quotes literature describing the interaction of Fe^0 with H_2 , showing barrier-less H-H bond splitting to give H-radicals that dissolve in Fe^0 . This can facilitate formation of BaH_2 from Ba^0 and H_2 (which normally is a high T process!). **Both processes seem highly relevant to the catalyst in question.** In this context, we forgot to mention that our finely divided Ba^0 alone already reacts with H_2 at room temperature to a Ba^0/BaH_2 mixture. Comprehensive experimental proof was supplied in our earlier paper on Ba^0 catalysis (*Angew. Chem. Int. Ed.* **2021**, *60*, 4252; see ESI: 1.6 Stoichiometric Experiments). We now included this statement in our revised version.

Literature quoted under (B) and (C) relates to the reaction of Ba hydride or Ba^0 with alkenes. **Also these processes are highly relevant to the catalyst in question.**

Point (D) quotes literature that describes the interaction between the Fe^0 surface and an unsaturated substrate like alkene or benzene showing the synergistic bonding/backbonding model which activates these substrates. **Also this process is highly relevant to the catalyst in question.**

Point (E) quotes experimental proof for the last step in the cycle: hydrogenolysis of Ba-alkyls with H_2 to give alkanes. We agree with the reviewer that there may be an alternative. Instead of product formation from Ba-alkyl and H_2 (σ -bond metathesis), one could also picture reaction of Ba-alkyl and H-radicals at the Fe^0 surface. We now included this possibility in the revised version.

Point (F) quotes older literature which show that there is an equilibrium between Ae^0/H_2 and Ae^0/AeH_2 (Ae = Ca or Ba). **Also this process is highly relevant to the catalyst in question.** We now added a reference to Pedersen who claims that $\text{Ba}^0 + \text{H}_2$ always gives substoichiometric $\text{BaH}_{<2}$, a mixture of Ba^0 and BaH_2 . In the ESI we describe evidence for presence of Ba^0 and Ba hydrides in the spent catalyst.

Point (G) quotes literature on the Haber-Bosch catalyst which describes the promoting effects of K. Although there is quite some discussion on the state of the K metal (K^+ or K^0) there is increasing evidence that the promotor is metallic K. Very recently, the group around Ferdi Schüth reported a Fe^0/Cs^0 catalyst for room temperature Haber-Bosch $\text{N}_2 \rightarrow \text{NH}_3$ conversion (*Angew. Chem. Int. Ed.* **2021**, *60*, 26385). Quote: "We tentatively attribute the promoting effect to the high electron donating

capability of the heavy alkali metals, similar to electron-donating effects observed in alternative systems.” Since we deal with electropositive Ba⁰ on Fe⁰, **also this process is highly relevant to the catalyst in question.** In fact, in 2001 Muhler showed that Ba-promoted Ru is a more effective Haber-Bosch catalyst than Cs-promoted Ru (*Angew. Chem. Int. Ed.* **2001**, *40*, 1061). We updated our discussion with the Muhler reference and the very recent work by Ferdi Schüth.

To summarize, we quoted literature that either deals with Fe⁰/H₂ or Fe⁰/substrate interactions or literature that deals with the Ba⁰/BaH₂ couple and its known reactivity. Although individual steps, previously reported knowledge is certainly useful for formulating a tentative mechanism.

As a result, the proposal in its current form is a collection of options with no way of limiting their number. This somewhat limits the usefulness of analysis and justification for its inclusion.

It may seem like a lot of possible pathways but in fact there are only two different pathways. (1) The most reactive metal (Ba⁰) reacts either first with H₂ (or H-radicals supplied by Fe⁰) or (2) first with the other reactant, the alkene. Both routes come together and then follow the classical cycle for alkene hydrogenation.

Many catalysis publications only report substrate scope and do not discuss any mechanism. As organometallic chemists who like to understand why things happen, we prefer to propose a mechanism.

The proposed mechanism is based on observations, combined with fundamental knowledge on chemical reactivity. We now found new, strong evidence for cooperation between a soluble main group component and a heterogeneous Fe⁰ component which is included in the paper (see below). We also completely revised the discussion on the mechanism and provide a new figure for the mechanism.

I invite the authors to perform additional experiments to clarify what makes this excellent catalytic system so special. Specifically, authors might opt at performing reaction kinetics measurements and identify the RDS in Ba vs BaFe catalyzed reduction. Similarly, comparative KIE analysis might explain why combining Ba and Fe gives the performance such a boost.

We agree that this is an excellent catalytic system. It is not difficult to see that it also holds a high potential for other catalytic conversions (which currently are under investigation). We also agree that, although our mechanistic proposal makes fully sense, conclusive evidence is extremely hard to obtain for a catalyst which is partially heterogeneous and magnetic, both precluding accurate NMR measurements.

Note that extensive experimental studies by Somorjai and Ertl “only” provided qualitative insights in the mechanism of Haber-Bosch N₂ → NH₃ conversion. More detailed insight was obtained by calculation and this is even for small molecules (N₂, H₂, NH₃) for heterogeneous systems an extreme challenge (see: “The transition state in heterogeneous catalysis” *Topics in Catalysis* **2002**, *18*, 141). The full Haber-Bosch mechanism was only very recently calculated (Goddard *et al. J. Am. Chem. Soc.* **2018**, *140*, 6288).

Experienced theoreticians here in Erlangen can calculate the interaction between Fe⁰ and benzene but cannot optimize a transition state for BaH₂ attack on benzene@Fe⁰. The latter system has considerably more degrees of freedom than the dissociation of N₂ or H₂ on a Fe⁰ surface. Since there are already several very good papers on calculations of benzene on a Fe⁰ surface, we simply refer in our paper to

published work. These papers show that the C-C bond in benzene is activated by Fe⁰ coordination, an observation that could be highly relevant to its accelerating effect in hydrogenation catalysis. In fact, it is generally agreed that *“in the hydrogenation over a group 8 metal catalyst, benzene is associatively adsorbed, probably as a π -complex”* (this statement comes directly from an older, authoritative review on aromatic hydrogenation: A. Stanislaus, B. H. Cooper, *Catal. Rev. Sci. Eng.* **1994**, *36*, 75) which we now also cite in the revised version.

The reviewer proposes kinetic measurements which could identify the rate-determining-step (RDS). In heterogeneous catalysis, such studies are complicated by transport effects and can only be modelled by Langmuir-Hinshelwood which already for simple systems is quite complicated. Such measurements may give some support to a mechanism but will never fully prove it.

However, following the reviewer's advice, we performed simple kinetic studies that show the dependence of the conversion rate on either H₂ pressure or substrate concentration. It was found that the H₂ pressure has essentially no effect (Table S4). We now also include observations on the effect of the substrate (alkene or arene) concentration (Table S5) and show that the conversion rate is independent on concentration. Although these measurements are only qualitative, the observed pseudo-zero order behavior of both reactants (H₂ and substrate) is typical for a heterogeneous reaction in which substrate-active site interaction is the rate-limiting factor; *i.e.* the bottleneck of the reaction is a heterogeneous step. We think this insight strongly supports our proposed mechanism, especially the step in which the substrate is bound to the Fe⁰-surface.

Considering analysis of the kinetic isotope effect (KIE), we are a bit more skeptical. Morris Bullock writes: *“While such measurements may be relatively simple to carry out experimentally, they are seldom unambiguously interpretable, since many steps contribute to the overall catalytic cycle.”* (Encyclopedia of Catalysis). John Hartwig writes: *“However, the interpretation of a KIE is not as simple as the measurement of a KIE.”* (*Angew. Chem. Int. Ed.* **2012**, *51*, 3066). We fully agree with both authors. Although we can easily repeat catalysis with D₂ instead of H₂, it is not clear what the isotope effect will learn us about the mechanism. This is especially in heterogeneous catalysis not easy.

In fact, we did repeat alkene hydrogenation with BaFe using D₂ instead of H₂ and found considerably faster conversion (at least by a factor 2). This means that there is an inverse isotope effect, at least under the chosen conditions. Interestingly, also in Fischer-Tropsch a negative isotope effect is found (*J. Phys. Chem. C* **2010**, *114*, 19761), however, since not only kinetic but also thermodynamic effects can contribute to the isotope effect, this can change with temperature. Kinetic isotope effects for CO hydrogenation on Ni have been reported but with contradicting conclusions (*J. Catal.* **1979**, *60*, 169). For this reason, we prefer not to discuss the observed inverse isotope effect with BaFe. This would bring even more confusion!

Below are some follow up points on the mechanistic part:

1) Line 194: The claim is not correct or valid in isolation. The fact that addition of soluble barium instead of metallic one promotes catalysis is not relevant to the catalytic system in question. It doesn't prove that homogeneous species are in fact produced when using Ba/Fe, at least not in current wording.

We agree with the reviewer. The fact that the catalytic activity of soluble Ba hydride clusters like $[\text{Ba}(\text{H})\text{N}''']_7$ is boosted by addition of Fe^0 does not prove that the Ba^0/Fe^0 system produces soluble Ba hydride species. It is not entirely impossible that soluble $[\text{Ba}(\text{H})\text{N}''']_7$ during catalysis converts to Ba^0 . Homogeneous catalysts can turn out to be heterogeneous and *vice versa*. Due to this uncertainty we rephrased the sentence:

“The fate of the soluble $[\text{Ba}(\text{H})\text{N}''']_7$ cluster under catalytic conditions is unclear. Therefore, this is not ultimate proof for a soluble main group metal component.”

The good news is that we now provide convincing evidence for a catalyst system with a soluble main group metal component and insoluble Fe^0 .

The soluble, very robust, Mg hydride complex $[(\text{BDI})\text{MgH}]_2$ alone is not catalytically active in arene hydrogenation. However, in the presence of Fe^0 , benzene is fully converted to cyclohexane within 2 hours (1.5 mol%, 150 °C, 50 bar H_2) and therefore at least as active as BaFe. After catalysis the Fe^0 can be separated with a magnet and the mother liquor clearly contains intact $[(\text{BDI})\text{MgH}]_2$. We believe that this is very strong evidence for a mechanism in which the soluble main group catalyst works in concert with a heterogeneous Fe^0 catalyst. We included these exciting new results in the paper and in the ESI.

2) Lines 207-208: This is very impressive.

We agree that benzene reduction with $\text{LiAlH}_4/\text{Fe}^0$ is very impressive. LiAlH_4 or Fe^0 alone are fully inactive in benzene hydrogenation. However, looking at Table 1, it should be realized that the Ba^0/Fe^0 system (TOF >33) is considerably more active than $\text{LiAlH}_4/\text{Fe}^0$ (TOF = 9) and therefore even more impressive!

3) Point A: Could authors clarify the state of hydrogen in each case - atomic vs hydride?

This is a fair point. The two metals Ba^0 and Fe^0 react with H_2 in completely different ways. Like most transition metals, Fe^0 with a relatively high electronegativity of 1.83 splits H_2 in atomistic H which can migrate freely in the Fe crystal lattice occupying tetrahedral or octahedral interstitial sites. In contrast, Ba^0 with a low electronegativity of 0.89, reacts with hydrogen to produce Ba^{2+} cations and H^- anions which form a salt.

In short, we included this information in the paper.

3) Point C is not demonstrated experimentally in the context of this work

It is indeed true that in the paper's original form, we did not supply proof for the reaction of Ba^0 and $\text{Ph}_2\text{C}=\text{CPh}_2$. We only referred to the reaction of Ba^0 with activated alkenes quoting Anwander and Mashima (now reference [34]). However, in our own previous work (now ref [26]), we describe in the ESI that reaction of Ba^0 and $\text{Ph}_2\text{C}=\text{CPh}_2$ gave a dark-red solution of $[\text{Ba}^{2+}][\text{Ph}_2\text{C}-\text{CPh}_2]^{2-}$ which after quenching with CD_3OD becomes colorless and gave $\text{Ph}_2\text{C}(\text{D})-\text{C}(\text{D})\text{Ph}_2$. We now include this information.

4) Point D demonstrates that Fe can promote LAH reduction (this was noted by other referees) but is does not have direct relevance to the Fe-Ba catalyst in question

We agree and thank the reviewer for this insight. The BaFe and LiAlH₄/Fe are completely different systems. This was not immediately clear to us. The problem with LiAlH₄ is that it is meta-stable and at 150 °C it decomposes to Li₃AlH₆, Al⁰ and H₂. This complicates the catalytic experiment and for that reason we took these experiments out and focus only on Fe/s-block metal systems which are considerably more active. Additional investigations on the Fe/Al system are in progress.

5) Point F is much appreciated. This is an exemplary, thoroughly analyzed piece of data that is directly relevant to catalysis in question.

We agree that this is compelling evidence for a soluble main group metal component. The fact that before catalysis there is a Ba/Fe ratio of 1/1 and after catalysis 2/1 (the surface is rich in Ba) shows that Ba goes in solution during catalysis and precipitates on the surface after catalysis (either as Ba⁰ or BaH₂).

B) State and composition of the catalyst:

1) Elemental composition: can authors truly call their samples bare? EA evidences significant amount of carbon and nitrogen in extent of few percent and no mention of that is given in XPS data analysis. What is the role of these components? Do they affect catalysis?

This is a fair point. Although X-ray tells us that we deal with microcrystalline Fe⁰ and Ba⁰, there is indeed C, N and O (and other elements, Si?) on the surface (and possibly also on the inside of the particles). These are remnants of cocondensation with organic solvents. This does not mean that these powders are not metallic. Powder diffraction and XPS confirm Fe⁰ and Ba⁰. The Fe⁰ powder is magnetic. The Ba⁰ powder reacts as Ba metal. Note that Ba⁰ has a very high N-content (>3%). The high affinity of Ba for N₂ is well-known and one of the reasons that it is the ideal metal to be used as a gas getter in radio tubes. The high N-values are due to molecular N₂ bound on the Ba⁰ surface (see: M.D. Malev, Gas sorption by barium films, Vacuum 1973, 23, 359) and not nitride which needs much higher temperatures for formation. In that sense, the metals surfaces are indeed not bare. We deleted the word "bare" and also indicate N now in the XPS Figure in the ESI.

Whether surface-bound C, H or N impacts the catalytic activity cannot be easily answered. We are not able to make these catalysts without these impurities. For N₂ adsorbed on the Ba⁰ surface it is known that it can be replaced by H₂, so this won't affect catalysis (See: "The clean-up of various gases by magnesium, calcium, and barium" A. L. Reimann, *Philosophical Magazine Series 1* **1934**, 18, 1117).

A

2) Mechanical agitation: This is a very interesting observation and authors might follow up on why mechanical treatment impacts the catalytic activity. It is very unexpected and i urge you to address this.

Firstly, this is critical for reproducibility. Secondly, does the electronic structure of the mixed catalyst change upon mechanical treatment?

The procedure of grinding the two metals is a primitive way of mixing metal powders but is reproducible. I would like to underscore that grinding does not make smaller particles or forms a legation. As we wrote earlier in the ESI, Ba and Fe do not form alloys (we now included this statement also in the main paper). XPS and powder X-ray diffraction spectra for the ground BaFe mixture are superpositions of the spectra for the pure Ba and Fe metals. The origin of the increased activity must be due to intimate contact between the two metals and is not due to changes in electronic structure. Since Ba is rather soft (1.25 Mohs) and Fe much harder (4.0 Mohs), grinding indeed could improve surface contact. We also noticed that a Ba/Fe mixture can be separated magnetically but a ground Ba/Fe mixture not. We included this now in the paper.

3) Authors are encouraged to be very specific with respect to their numbers. In the current SI it is very unclear which amounts of catalysts were used especially in mixed catalyst systems and ones with alkali metal hydrides. In some instances the numbers are simply not there.

We agree with the reviewer that the method of alkene hydrogenation has been fully described for the BaFe catalyst, but lacks in details for the Fe⁰/X combination (X = s-block metal species). The exact numbers of weighed-in substrate, Fe⁰ and cocatalyst X can now be found in the ESI.

C) Further remarks in order of occurrence:

1) Can the authors clarify the LAH and DIBAL data? These are significantly different from Ae co-catalysts and might operate via a different mechanism, e.g. they show activity in the absence of H₂. In current state it appears you cannot reliably group these with Ae species. A control experiment would help clarifying this - e.g. one in the absence of H₂ and one without aqueous workup.

We fully agree with the reviewer that these cocatalysts are quite different from s-block metal cocatalysts (see our statement (4) above). This is why we took the Fe/Al systems out. They were clearly inferior to BaFe anyway.

2) lines 35-40 - a more exhaustive description of homogeneous catalysis for alkene hydrogenation is needed for a broad readership journal

This remark is related to point 4) below and we like to treat this together. The reason why we show both general cycles is simple: all s-block metal catalysis is based on non-redox cycles. We like to stress this point. The reviewer correctly noticed that hydrogenation catalysis is much more diverse. We now extended the introduction with references to different concepts for alkene hydrogenation (cooperative systems, CHAT, FLP, photoactivation etc....). Equally important, we also extended the introduction with a more in-depth discussion of our earlier reported Ba⁰ catalyst. In order to fully understand the

mechanistic discussion for the BaFe catalyst, first our previous paper on Ba⁰ catalysis must be understood.

3) Figure 1 - a reference to the reactivity order is missing

This has now been included.

4) Lines 47-50 - while i agree with the introduction of two mechanisms, the examples of redox non-active processes are very limited and somewhat insular. There are a lot of homogeneous (cooperative/bifunctional) catalyst that operate without redox and are not mentioned

See point 2) above.

5) Authors might reconsider their amide notation. The N'' is actually HMDS - an exhaustive 4-letter notation that is clear to the majority of organic chemists

HDMS and N'' are both frequently in use. For consistency with our earlier papers on this amide ligand (there are many!) we prefer to stick to N''. There is also a practical reason for the N'' notation: it is much easier to draw the complicated [Ba(H)N'']₇ cluster in (now) Fig. 2b using N'' instead of HDMS. As long as we define the abbreviation, it should be ok.

6) Lines 67-68: Could you elaborate on the nature of BaH₂ and give the reference to the mentioned reactivity?

This is a good point that may need to be elaborated. Pedersen showed that, in reaction with H₂, Ba metal is never completely converted to BaH₂. Similar as known for transition metals, a substoichiometric product is obtained (BaH_{1.2}). This mixture of Ba⁰ and BaH₂ has been proposed to be the key to its unexpected high activity in catalysis (see the references to Wright and Weller – now 36-38). We now included the reference to Pedersen on substoichiometric BaH_{1.2} (ref. 33).

7) Line 119: Could you elaborate on your use of the term "activated"?

We mean here activated by evaporation/cocondensation, i.e. the MVS (metal vapor synthesis) method. We now defined MVS in the introduction and refer to this powder as being MVS-activated.

8) Figure 3 and the manuscript: As long as there is no kinetic data, authors cannot use the term TOF, which is a kinetic performance parameter and cannot be estimated from single point. This would be strongly misleading and unjustified.

First of all, it should be noted that in most cases we give TOF's for 99%, more or less full, conversion (we aim at full conversion to demonstrate the applicability of our method). In contrast, most research articles give TOF's for the initial conversion (5-20% conversion) which are generally much higher.

We do not agree that TOF's cannot be estimated from a single point. The TOF is simply the TON, which can be estimated at any single point, divided by time. However, we do agree with the reviewer that TOF's for reactions with full conversion can be meaningless in case it is not clear when full conversion was reached.

Maybe this wasn't clear but in our investigations the times for full conversion have always been optimized. Our TOF estimations for full conversion are therefore not based on a "single point" observation. The time for 99% conversion has been optimized by doing many experiments with different reaction times. This was written in the caption of Figure 3 and was also described in the Supporting information but may not have been obvious. We now elaborated this description in the Supporting Information. For fast conversion (< 1 h) we optimized the time in 0.1 h steps. For slower conversions we took larger steps. In cases of very long reaction times (24 h) we now reinvestigated some entries for 99% conversion. We found that naphthalene and imine convert slightly faster than 24 h (22 h and 23 h, respectively). This has been corrected in Figure 3 but has hardly any effect on the TOF and on our conclusions.

Note that for very sluggish reactions (with only-Ba⁰ or only-Fe⁰ catalysts) TOF's are given for incomplete conversion. If we would strive for full conversion (99%), we would need extremely long reaction times! The TOF's would be even lower and the difference with the BaFe catalyst would be even more extreme.

Although we agree that a full kinetic investigation (*k* rate constants) would be the most accurate way of comparing catalyst activities, we feel that the use of TOF gives a very good approximation of differences in catalyst activity. Using TOF this way is standard in many papers dealing with catalysis.

9) Figure 3 caption: typo in the word "shut"

Corrected.

10) Line 167 and throughout the manuscript: Authors need to clarify the weight vs surface aspect of their catalytic system. Since the catalyst is heterogeneous, how do weight rations translate to surface ones? This point appears to be important since even mechanical treatment of pristine powder mixtures does strongly affect the performance (Line 170)

First of all, simple mechanical treatment with mortar and pestle is not going to change the size of our (nano)particles. Powder X-ray measurements confirm that Ba⁰, Fe⁰ and the ground BaFe mixture are all microcrystalline particles of approximately 5 nm. We think that mechanical treatment gives a better surface contact. See point A(2) above.

We agree with the reviewer that our catalysts are far more active than stated in the paper. TON and TOF values are given as conversion of substrate molecule per catalyst metal atom. Since the Fe part of the catalyst is fully heterogeneous, only surface atoms can be active. For ideal spherical particles of 5 nm (which was estimated for our MVS-Fe⁰), the degree of dispersion (ratio of surface to total atoms) is circa 15-20% (see: G. C. Bond, *Metal Catalysed Reactions of Hydrocarbons*, Springer, **2005**). This means that, calculated per metal surface atom, our catalysts are at least a factor 5-6 times more active. Since particle aggregation and non-ideal shapes could change this factor, measurements of the active surface could give a more accurate estimation. Having said that, this also would only be a rough approximation.

For an exact indication of catalytic activity, knowledge on the activities of specific crystal faces and the role of lattice defects would be needed.

We agree the subject is important but also strongly feel that in this preliminary stage of the project, any detailed discussion on this subject leads to more confusion. For now, we simply included a remark on catalyst activity/weight *versus* catalyst activity/surface atoms in the Supporting Information.

11) Line 179:

We are not sure what is needed here but assume this is connected to point 3 below.

SI:

1) The accuracy of MVS synthesis description is critical for the reproducibility of results. Currently this description contains, i quote: "rule of thumb". How are other researchers supposed to reproduce this?

"Rule of thumb" relates to the desired temperature for metal evaporation. In a typical experiment the temperature is slowly raised and as soon as the metal starts to melt we usually see metal precipitation on the outer flask walls. At this point we measure the temperature of the metal container. To maintain a reasonable flow of metal vapor, the operation temperature is set circa 100 °C higher. It is clear that this temperature also strongly depends on the local vacuum which may vary from machine to machine. We understand that the reviewer wants to see accurate temperatures. We now give for each metal an accurate temperature (+ pressure) but also give a warning that this is strongly vacuum and machine dependent.

2) The protocol for Fe oxalate decomposition is missing and should be described

This is a good point. The nature of decomposition is very much dependent on the atmosphere (air, inert gas, H₂). We decomposed Fe oxalate under vacuum by heating a 25 ml Schlenk flask with 2.0 g of yellow Fe-oxalate·(H₂O)₂ with a Bunsen burner to circa 400 °C until CO₂ release stopped (after circa 10 minutes). Due to fast CO₂ evaporation, considerable quantities of pyrophoric iron are lost in the tubing and the dust filter. The yield of pitchblack pyrophoric iron was 1.0 g. According to literature this is an approximately equimolar mixture of Fe⁰ and Fe₃O₄. For this reason, we used in catalysis experiments the double amount of pyrophoric iron. We describe this procedure now in the Supporting Information.

3) SI and line 179 - the data on elemental analysis of spent catalytic medium is not included. Without it and adequate detection limit description the claim on "quantitative" removal is unsupported.

We show in Figure S16 in the Supporting information the removal of the BaFe catalyst after benzene hydrogenation. The last photo in the series is an empty flask obtained after evaporating all volatiles from the mother liquor from which the catalyst (in this particular case ca. 100 mg!) was removed magnetically. We intended to do elemental analysis on the remains after evaporation. However, looking at the flask, we could visually not detect any remains, not even with a microscope! Also after upscaling,

using a larger quantity of BaFe catalyst (1.0 g), no visible quantities of catalyst remain in the mother liquor. Since it makes no sense to hand in an empty flask for elemental analysis, we did not proceed further with this idea. Although we cannot rule out loss of trace ($< \mu\text{g}$) quantities of metal, this is to our opinion “quantitative removal”. We do agree that others may have a different definition of the word “quantitative” and therefore we simply deleted it in the manuscript and now write: from which the catalyst can be ~~quantitatively~~ removed with a magnet and reused without significant loss of activity (Fig. 4). We also updated this section in the Supporting Information.

We now included an additional elemental analysis of the spent BaFe catalyst which shows the expected Ba/Fe ratio of circa 1/1 and the presence of some organics (C and H).

Summary: I find the results very appealing and invite the authors to act on the comments above. Once these issues are resolved, I'd be happy to recommend the publication.

We agree that the results of this contribution are clearly very appealing. We suspect that this paper will have very high visibility and that our approach will also find followers.

REVIEWERS' COMMENTS

Reviewer #2 (Remarks to the Author):

Authors have submitted their revised manuscript with some corrections and a reply to my comments.

Most of the comments were addresses and the additions seem appropriate.

I understand their points on mechanistic analysis and while at times I disagree with authors strongly this disagreement does not principally diminish the value of work which is interesting and would have value for community in my opinion.

Some final points:

1) I do not agree with the treatment of mechanistic proposal and will reiterate the point -

If a something makes sense it does not mean that it actually takes place.

Authors collect all relevant literature data for Ba, Fe et al, place it on the figure and refer to it as a "working hypothesis". This activity has no value - it does not prove anything but summarizes what can "hypothetically" happen.

The work is good enough on its own and a crude mechanistic proposal with limited support is just unnecessary. This is my opinion.

2) I repeat my previous comment regarding the use of TOF and TON values. In their reply authors mentioned "The TOF is simply the TON, which can be estimated at any single point, divided by time."

This is plain wrong.

Mathematically TOF is an instantaneous value and treating it like this is bad practice that has neither meaning nor justification in the context of this work.

Authors are invited to read the viewpoint on that (<https://pubs.acs.org/doi/10.1021/cs3005264>) and have a look at examples discussed.

Reviewer #2:

Authors have submitted their revised manuscript with some corrections and a reply to my comments. Most of the comments were addresses and the additions seem appropriate.

I understand their points on mechanistic analysis and while at times I disagree with authors strongly this disagreement does not principally diminish the value of work which is interesting and would have value for community in my opinion.

Some final points:

1) I do not agree with the treatment of mechanistic proposal and will reiterate the point - If a something makes sense it does not mean that it actually takes place.

Authors collect all relevant literature data for Ba, Fe et al, place it on the figure and refer to it as a "working hypothesis". This activity has no value - it does not prove anything but summarizes what can "hypothetically" happen.

The work is good enough on its own and a crude mechanistic proposal with limited support is just unnecessary. This is my opinion.

We appreciate and respect the opinion of reviewer 2 and most important thank the reviewer for the time and effort spend on reviewing our paper. The reviewer's insight in catalysis and critical remarks has certainly contributed considerably to the quality of its final version. We feel the main problem is the interplay between homogeneous and heterogeneous catalysis – which is quite unusual. We spend a lot of time in finding a convincing example in which a soluble catalyst, the [(BDI)MgH]₂ complex in Fig. 5, works in synergy with the Fe⁰ surface and returns after catalysis in its original form. Assuming that Fe does not go in solution (we have indications for this), catalysis MUST proceed at the interface of solution and the Fe⁰ surface. For this reason, the following comment is not completely fair: "Authors collect all relevant literature data for Ba, Fe et al, place it on the figure and refer to it as a "working hypothesis". We not only refer to literature but have various experimental observations (among others, the fact that Ba does go into solution during catalysis but Fe not) that support our mechanism. It is the combination of experiment, literature and common knowledge that led to the postulated mechanism.

At the other hand, we also feel the frustration not to be able to fully prove each intricate detail of the mechanism. A full kinetic analysis would not provide us with this either. The only convincing way is probably computer modelling. Many calculations on Fe-substrate interactions have already been published (we refer to them) but transition states and E-profiles are not available. We are further looking into possibilities. If possible, and this is not guaranteed, this will be a substantial calculational project.

ACTION: To underscore our latest evidence for a mechanism in which a soluble metal hydride complex cooperates with a solid Fe⁰ surface, we added to the text: "Based on these observations, it is likely that catalysis proceeds at the solid-solution interface." We also added in the final conclusion the following sentence: "While we are further scrutinizing the intricate details of this tentative mechanism, the hitherto made observations have important ramifications for future research." We strongly believe that

the here observed “hybrid-catalysis”, interplay of homogeneous and heterogeneous catalysis, has a bright future. We are working full force on further development of this approach.

2) I repeat my previous comment regarding the use of TOF and TON values. In their reply authors mentioned "The TOF is simply the TON, which can be estimated at any single point, divided by time."

This is plain wrong.

Mathematically TOF is an instantaneous value and treating it like this is bad practice that has neither meaning nor justification in the context of this work.

Authors are invited to read the viewpoint on that (<https://pubs.acs.org/doi/10.1021/cs3005264>) and have a look at examples discussed.

We are fully aware of Kozuch’s paper on TOF/TON issues. In fact, I am a big fan of his work and also personally use Kozuch’s Energetic-Span-Model in calculation studies.

In my last rebuttal I indeed wrote: “The TOF is simply the TON, which can be estimated at any single point, divided by time.” This is what it is. I do not know any other definition (apart from the different TOF values discussed by Kozuch). We fully agree with the reviewer that TOF is an instantaneous value and therefore time dependent. This is one of the shortcomings discussed in Kozuch’s paper.

The reviewer must have noticed that in our rebuttal immediately following our quote “The TOF is simply the TON, which can be estimated at any single point, divided by time”, we continue with: “However, we do agree with the reviewer that TOF’s for reactions with full conversion can be meaningless in case it is not clear when full conversion was reached.” This means that we acknowledge the factor “time”.

We are fully aware of the fact that fantastic TOF values can be reached at the beginning of the reaction (especially if there is no induction period). We are not looking for fantastic TOF values but for a fair comparison between catalyst systems. This is why (in most cases) TOF’s are given for full (99%) conversion AND, equally important, the times for full conversion have been optimized. This means we are not looking at a single time point but at several time points (as previously mentioned in our rebuttal). Note that TOF is also dependent on temperature and substrate concentration (see Kozuch paper). For this reason, we compare the Ba and BaFe catalysts at the same T(°C) and same concentration (in most cases we work solvent-free). We are convinced that this method allows for a fair comparison of catalyst activities, especially because the difference in activities is huge (often 3 orders of magnitude)! We would agree with the reviewer that small differences in TOF values (e.g. 20 h⁻¹ vs. 25 h⁻¹) would be meaningless.

ACTION: In the re-revised version, we added a note in the caption of Figure 3. We refer to the Supplementary Information for a more detailed discussion of TOF values and also included the reference to the Kozuch paper in this discussion.

Finally, I like to refer to a published comment of Kozuch: <https://pubs.acs.org/doi/10.1021/cs4000415>

which he starts with: “My favorite Chinese proverb says: Were I to await perfection, my book would never be finished”.

We feel this wisdom can also be applied to our current submission. We agree that, although there are very strong indications, the intricate details of our proposed mechanism are not fully proven. This may take a very long time – if possible at all. But is this not how science works? Something is proposed based on observation, literature and common knowledge. Anyone at any time can disprove our working hypothesis or report additional proof. As mentioned in the manuscript, we hope to isolate fully homogeneous heterobimetallic catalysts which will make it easier to investigate these intriguing systems.

Editorial changes:

We followed the checklist carefully and implemented the changes.

Other changes:

We changed a few words for better readability and found some typos which were corrected.